# Relationship between brain volume reduction during the acute phase of sepsis and activities of daily living in elderly patients: A prospective cohort study

**Toru Hosokawa, Kosaku Kinoshita\*, Shingo Ihara, Katsuhiro Nakagawa, Umefumi Iguchi, Marina Hirabayashi, Tomokazu Mutoh, Nami Sawada, Tsukasa Kuwana, Junko Yamaguchi**

Department of Acute Medicine, Division of Emergency and Critical Care Medicine, Nihon University School of Medicine, Tokyo, Japan

\* kinoshita.kosaku@nihon-u.ac.jp

## Abstract

Brain damage in acute sepsis may be associated with poor long-term outcomes that impair reintegration into society. We aimed to clarify whether brain volume reduction occurs during the acute phase of sepsis in patients with acute brain damage. In this prospective, noninterventional observational study, brain volume reduction was evaluated by comparing head computed tomography findings at admission with those obtained during hospitalization. We examined the association between brain volume reduction and performance of the activities of daily living in 85 consecutive patients (mean age, 77 ± 12.7 years) with sepsis or septic shock. The bicaudate ratio increased in 38/58 (65.5%) patients, Evans index increased in 35/58 (60.3%) patients, and brain volume by volumetry decreased in 46/58 (79.3%) patients from the first to the second measurement, with significant increases in the bicaudate ratio ($P < 0.0001$) and Evans index ($P = 0.0005$) and a significant decrease in the brain volume by volumetry ($P < 0.0001$). The change rate for brain volume by volumetry was significantly correlated with the Katz index ($ρ = −0.3790$, $P = 0.0094$). In the acute phase of sepsis in this sample of older patients, 60–79% of patients showed decreased brain volumes. This was associated with a decreased capacity for performing activities of daily living.

## Introduction

Sepsis is the leading cause of admission to the intensive care unit (ICU) [1]. Although the number of patients with sepsis is steadily increasing [2], the number of patients who survive and are discharged from the ICU has also increased due to technological advances in intensive care [3]. However, the quality of life of these patients after ICU discharge is lower than that of healthy people of a similar age [4]. Long-term outcomes are poor, especially in severe cases, and the survival rate after septic shock is approximately 40% [5]. In particular, sepsis-associated encephalopathy associated with brain dysfunction has been regarded as an important pathological condition related not only to survival rates but also to reintegration of patients into society [6,7].

**Data Availability Statement:** All relevant data are within the manuscript and its Supporting Information files.

**Funding:** The author(s) received no specific funding for this work.

**Competing interests:** The authors have declared that no competing interests exist.

In recent years, early intervention in diagnosis and treatment has been promoted, aimed at improving sepsis outcomes, the survival rate of severe sepsis [3], and long-term outcomes such as cognitive impairment [8].

To this end, we hypothesized that brain damage, including brain volume reduction that occurs in the acute phase of sepsis, may be associated with poor long-term outcomes. Evaluation of ventricular enlargement is as important as the evaluation of memory impairment [9,10]; however, previous reports on organic brain damage, decreased brain volume, and ventricular enlargement after sepsis have only used computed tomography (CT) [11,12] or magnetic resonance imaging (MRI) [13,14] for evaluation at a single timepoint during inpatient treatment. Furthermore, there are no reports evaluating brain volume reduction or ventricular enlargement over time.

Therefore, this study aimed to clarify whether brain volume reduction occurs during the acute phase of sepsis by focusing on acute septic brain damage. Furthermore, we examined its clinical significance, especially the correlation between poor activities of daily living (ADL) function and brain volume reduction or ventricular enlargement in the acute phase of sepsis.

## Patients and methods

### Patients and protocol

This was a single-center prospective observational study conducted at the Nihon University Itabashi Hospital. This study was conducted in accordance with the tenets of the Declaration of Helsinki. Approval was obtained from the clinical research institutional review board of the Nihon University School of Medicine Itabashi Hospital (RK-170912-08), and written informed consent was obtained from each patient or their family prior to enrolment. Patients admitted to our ICU between March 2018 and March 2020 were included. All patients had a clinical diagnosis of sepsis and septic shock as defined by the 2016 international sepsis criteria (Sepsis-3) [15]. The exclusion criteria were as follows: age < 20 years, in-hospital onset of sepsis, transfer from another hospital and intervention before admission to our hospital, and a Katz score [16] ≤ 4 before admission, indicating poor ADL function. Patients with in-hospital onset of sepsis and those who underwent intervention before admission were excluded because their hemodynamics were stabilized with early fluid therapy during the acute phase of sepsis [15], making the evaluation of brain volume reduction difficult due to the increased brain water content after fluid therapy. Moreover, patients with a Katz score ≤4 before admission were excluded because poor ADL function before admission is generally related to that at discharge.

Vital signs at admission were recorded for all included patients. Blood tests were performed on admission (day 0) and on day 7, day 14, day 21, and day 28. These periodic tests were routinely performed to determine the course of sepsis treatment. We used these results to calculate the daily Sequential Organ Failure Assessment (SOFA) scores. Head CT was performed at the time of admission and at the time of symptom change or discharge, as appropriate for follow-up. Vital signs, CT images, and blood biochemical data were documented in medical records. The diagnosis of sepsis was based on vital signs and blood test findings on admission.

### Bacteriological assessment

Blood culture was performed at the time of admission for all patients. To confirm the diagnosis of infectious illness, blood and bacterial cultures of specimens taken from organs considered to be the focus of infection based on clinical findings, imaging findings, and laboratory data were assessed. Although all patients met the Sepsis-3 criteria and clinical criteria for infectious illness, a subgroup of patients with negative bacterial culture results were defined as cases of

unknown origin. The group classification was based on whether the bacterial cultures of specimens taken from the sites considered to be of septic focus were positive.

## Assessment of brain volume reduction

We compared the head CT findings at the time of admission with those obtained during the course of treatment. Using these images, we investigated brain volume reduction, evaluated using the bicaudate ratio (BCR) [11,17], Evans index (EI) [18], and volumetry (Vo) [19,20]. A decrease in brain volume was defined as an increase in BCR and EI or decrease in the brain volume by Vo in the subsequent head CT compared with those in the initial head CT. BCR, EI, and Vo were defined as follows: BCR is the width between the anterior horns of the bilateral lateral ventricles divided by that of the cerebrum at the same height at the caudate nucleus level. It is used to evaluate frontal lobe volume loss; therefore, an increase in BCR implies a decrease in frontal lobe volume. EI is the maximum width between the anterior horns of the bilateral lateral ventricles divided by that of the cranial lumen at the same height. EI is often used to evaluate ventriculomegaly, but it is also used to evaluate brain atrophy [12]. An increase in EI is associated with a decrease in brain volume. Vo was calculated using an image processing workstation, Ziostation (Ziosoft, Inc., Tokyo, Japan) on CT images. All head CT images were transferred to Ziostation, where they were reconstructed [20]. Subsequently, whole-brain images were extracted and adjusted with a window value of 25–35 Hounsfield units, and the ventricular volume was subtracted to calculate the brain volume. All measurements were performed by two physicians, and the patients were blinded to the measurements.

## Assessment of ADL

ADL were assessed at discharge or patient transfer and were evaluated with the Katz index. Of the six functions in the Katz index, patients with moderate or severe disability (i.e., requiring assistance with at least two of the following: bathing, dressing, toilet use, transferring, continence, and eating) were excluded from the analysis [21,22].

To examine the association between brain volume reduction and ADL, the entire study population, including patients with brain volume reduction, was analyzed. We also compared demographic characteristics and ADL in the groups with and without brain volume reduction. Patients' demographic characteristics were age, the Acute Physiology and Chronic Health Evaluation (APACHE) II score, the SOFA score, presence or absence of septic shock, length of ICU stay, length of intubation, presence or absence of disseminated intravascular coagulation (DIC), serum leukocyte and platelet count, and serum values for hemoglobin, hematocrit, albumin, total bilirubin, aspartate aminotransferase, alanine aminotransferase (ALT), sodium, potassium, urea nitrogen, creatine, lactate, bicarbonate, C-reactive protein, glucose, antithrombin 3, and uric acid. The scores and blood test values at admission were used for examination.

## Assessment of fluid balance

We investigated whether fluid balance affected the reduction in brain volume. The in-out fluid balance for 24 h after admission and on the day of the final CT was calculated and compared between groups with and without brain volume reduction.

## Statistical analysis

Statistical analysis was performed using JMP version 13 (SAS, Cary, North Carolina, USA). Collected measurements were analyzed for normal distribution using the Shapiro–Wilk test. A

*P*-value <0.05 was considered statistically significant. Discrete variables are indicated by integer values (%). In the case of continuous variables, normally distributed data are shown as mean ± standard deviation and non-normally distributed data as median and interquartile range. The Wilcoxon signed-rank test was performed for the comparison of two groups of paired nonparametric data. Spearman's rank correlation test was performed to evaluate correlations. The Mann–Whitney U test was performed for the comparison of the groups.

## Results

### Clinical features

Among the 137 patients with sepsis and septic shock admitted to our ICU between March 2018 and March 2020, consent was obtained from 112 patients. Patients whose Katz score was ≤4 before admission (n = 27) and whose brain CT could not be performed due to death (n = 19), hospital transfer (n = 2), department transfer (n = 2), discharge (n = 1), or other reasons (n = 3) were excluded. Thus, 58 patients were enrolled, and they underwent brain CT at two timepoints (Fig 1).

Clinical and demographic characteristics of the study population are presented in Table 1A. The mean age was 77 ± 12.7 years, 24 were women (41.4%), the APACHE II score at admission was 24 ± 7.0, SOFA score was 7 ± 3.2, and septic shock occurred in 17 (29.3%) patients. The length of ICU stay was 8 (5–12) days, length of hospital stay was 20 (12–32.25) days, and length of intubation was 2.5 (0–6) days. The mortality rate was 5%, tracheostomy was performed in 6

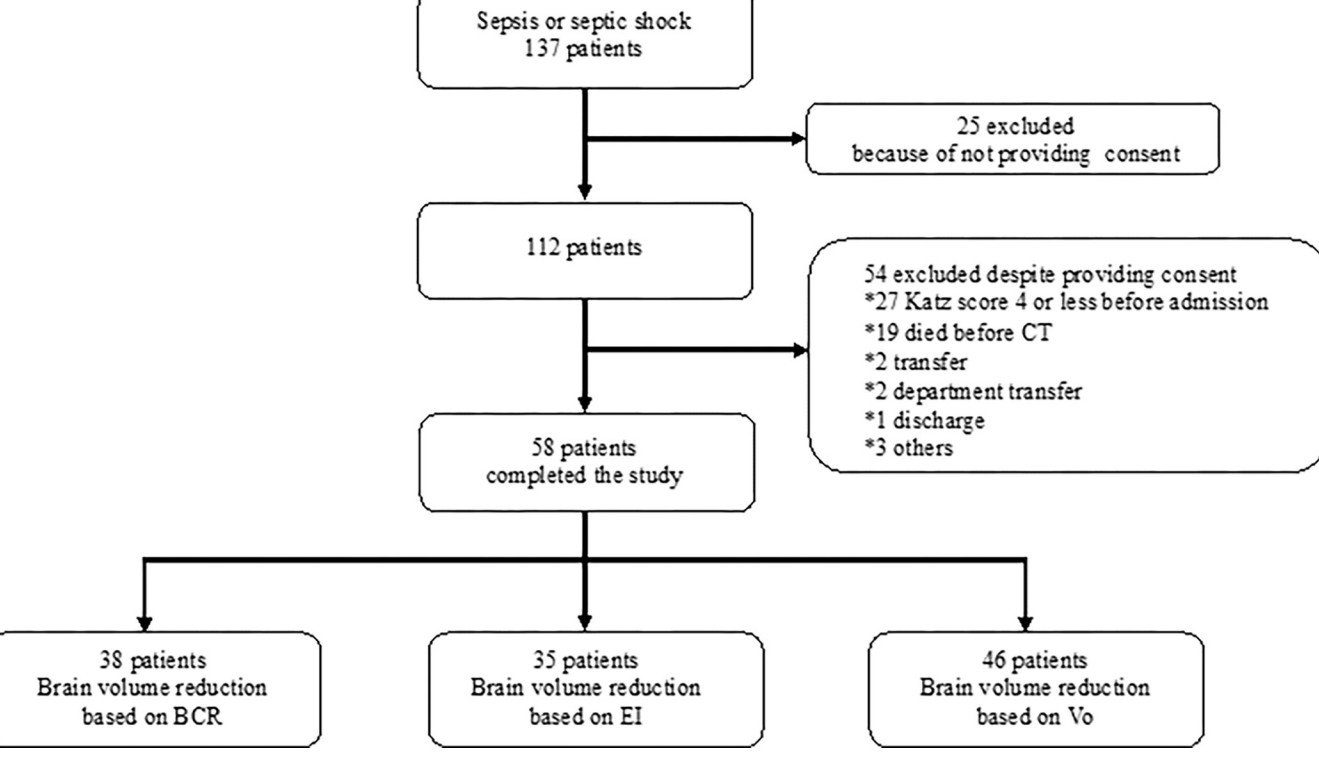

**Fig 1. Flowchart of the inclusion and exclusion criteria.** In total, 137 patients were diagnosed with sepsis or septic shock during the study period, of whom 112 patients provided informed consent for this study. Among these, patients with a Katz score ≤4 before admission, early death, hospital transfer, department transfer, and discharge were excluded; finally, 58 patients were included in this study; they underwent head CT at ≥2 timepoints, including the time of admission. In total, 38 patients had brain volume reduction based on the BCR, 35 patients had brain volume reduction based on the EI, and 46 patients had brain volume reduction based on Vo findings. Abbreviations: CT, computed tomography; BCR, bicaudate ratio; EI, Evans Index; Vo, volumetry.

**Table 1. Characteristics and outcomes of the study population.**

(A)

| Variables | n = 58 |
|---|---|
| Age (years) | 77 ± 12.7 |
| Female, n (%) | 24 (41.4) |
| APACHE II score at admission | 24 ± 7.0 |
| SOFA score at admission | 7 ± 3.2 |
| Septic shock, n (%) | 17 (29.3) |
| Days in the intensive care unit | 8 (5–12) |
| Duration of hospitalization (days) | 20 (12–32.25) |
| Duration of intubation (days) | 2.5 (0–6) |
| Mortality, n (%) | 5 (8.6) |
| Tracheotomy, n (%) | 6 (10.3) |
| Hemofiltration, n (%) | 10 (17.2) |
| DIC, n (%) | 25 (43.1) |
| Katz score at discharge | 1 (0–4) |
| Duration between the two CT scans (days) | 13 (7–17) |
| WBC ($\times 10^3$/μl) | 10.75 (7.375–15.975) |
| Hemoglobin (g/dl) | 13.15 (11.4–14.9) |
| Hematocrit (%) | 40.55 (34.4–43.5) |
| Platelet ($\times 10^4$/μl) | 18.2 (12.75–27.9) |
| Albumin (g/dl) | 3.3 (2.7–3.825) |
| T.Bilirubin (mg/dl) | 0.65 (0.415–1.0075) |
| AST (U/l) | 47 (24.75–85.5) |
| ALT (U/l) | 28.5 (14–67.75) |
| Na (mEq/l) | 138 (134.75–144) |
| K (mEq/l) | 4.3 (3.9–4.8) |
| BUN (mg/dl) | 36.4 (20.875–61.5) |
| Creatinine (mg/dl) | 1.51 (0.98–2.2825) |
| Lactate (mmol/l) | 2.85 (1.975–4.825) |
| $HCO_3^-$ (mmol/l) | 19.95 (15.925–23.5) |
| CRP (mg/dl) | 9.995 (1.6175–25.3175) |
| Blood glucose level (mg/dl) | 158.5 (104–229.75) |
| AT3 (%) | 79.5 (64–88.5) |
| UA (mg/dl) | 7.4 (5.5–9.95) |

(B)

| Site of Infection | n = 58 | Pathogen |
|---|---|---|
| Pulmonary, n (%) | 31 (53.4%) | |
| Pure gram negative, n (%) | 7 (12.1%) | *Escherichia coli, Haemophilus influenzae, Klebsiella pneumoniae, Pseudomonas aeruginosa* |
| Pure gram positive, n (%) | 8 (13.8%) | *Staphylococcus aureus, Streptococcus agalactiae* |
| Mixed bacteria, n (%) | 9 (15.5%) | *Klebsiella pneumoniae, Escherichia coli, Streptococcus pneumoniae, Streptococcus anginosus, Enterobacter cloacae* |
| Influenza virus, n (%) | 2 (3.4%) | |
| *Candida albicans*, n (%) | 1 (1.7%) | |
| No pathogen, n (%) | 4 (6.9%) | |
| Positive blood culture, n (%) | 3 (5.2%) | |
| Urinary tract, n (%) | 15 (25.9%) | |
| Pure gram negative, n (%) | 10 (17.3%) | *Escherichia coli, Klebsiella pneumoniae, Citrobacter koseri* |

*(Continued)*

| | | |
|---|---|---|
| Pure gram positive, n (%) | 3 (5.2%) | *Staphylococcus aureus, Staphylococcus epidermidis, Enterococcus raffinosus* |
| Mixed bacteria, n (%) | 1 (1.7%) | *Escherichia coli, Staphylococcus epidermidis* |
| No pathogen, n (%) | 1 (1.7%) | |
| Positive blood culture, n (%) | 7 (12.1%) | |
| Intra-abdominal, n (%) | 6 (10.3%) | |
| Pure gram negative, n (%) | 3 (5.2%) | *Escherichia coli, Pseudomonas aeruginosa* |
| Pure gram positive, n (%) | 1 (1.7%) | *Streptococcus gordonii* |
| Mixed bacteria, n (%) | 2 (3.4%) | *Klebsiella pneumoniae, Streptococcus faecalis, Citrobacter koseri* |
| Positive blood culture, n (%) | 3 (5.2%) | |
| Soft tissue, n (%) | 3 (5.2%) | |
| Pure gram positive, n (%) | 3 (5.2%) | *Streptococcus agalactiae, Staphylococcus aureus, Staphylococcus epidermidis* |
| Positive blood culture, n (%) | 2 (3.4%) | |
| Others, n (%) | 3 (5.2%) | |
| Pure gram positive, n (%) | 2 (3.4%) | *Staphylococcus aureus* |
| Mixed bacteria, n (%) | 1 (1.7%) | *Streptococcus faecalis, Streptococcus constellatus* |
| Positive blood culture, n (%) | 2 (3.4%) | |

Abbreviations: *ALT* alanine aminotransferase, *APACHE* Acute Physiology and Chronic Health Evaluation, *AST* aspartate aminotransferase, *AT* antithrombin, *BUN* blood urea nitrogen, *CRP* C-reactive protein, *DIC* disseminated intravascular coagulation, *SOFA* Sequential Organ Failure Assessment, *UA* uric acid, *WBC* white blood cell.

(10.3%) patients, and hemofiltration therapy for acute kidney injury was performed in 10 (17.2%) patients during hospitalization. DIC was observed in 25 (43.1%) patients during hospitalization. The mean ADL score at discharge or transfer was 1 (0–4) as per the Katz index. ADL function was evaluated at a median of 20 (12–32.25) days from admission. Original infections included pneumonia in 31 (53.4%) patients, urinary tract infection in 15 (25.9%), intra-abdominal infection in 6 (10.3%), soft tissue infection in 3 (5.2%), and others in 3 (5.2%) (Table 1B). The "other" category included one case each of periapical dental infection, infectious endocarditis, and bacteremia with unknown foci. Bacteremia was observed in 17 (29.3%) patients. The pathogens were pure gram negative in 20 (34.5%) patients, pure gram positive in 17 (29.3%), mixed bacterial in 13 (22.4%) patients, influenza virus in 2 (3.5%) patients, and *Candida albicans* in 1 (1.7%) patient. Pathogens could not be identified in 5 (8.6%) patients (Table 1B) because the specimens were insufficient for analysis in three cases and antimicrobial agents were administered before specimen collection in two cases.

Nonparametric data are presented as median with interquartile range. Categorical data are presented as n (%). The biochemical data in this table are based on data at the time of admission (i.e., at the time of sepsis diagnosis). The "other" category included one case each of periapical dental infection, infectious endocarditis, and bacteremia with unknown foci.

## Brain volume reduction

The mean timepoint at which the second head CT was performed was the 13th (interquartile range, 7–17) day of hospitalization. A comparison between the findings from the two timepoints of head CT showed a significant increase in BCR in 38/58 (65.5%, $P < 0.0001$) patients, significant increase in EI in 35/58 (60.3%, $P = 0.0005$), and significant decrease in Vo in 46/58 (79.3%, $P < 0.0001$) (Fig 2). Regarding the extent of brain volume changes, BCR increased by 5.9% (interquartile range, 0–15.7), EI increased by 1.4% (interquartile range, 0–3.6), and Vo decreased by 1.8% (interquartile range, 0.3–4.5) compared with the values on admission.

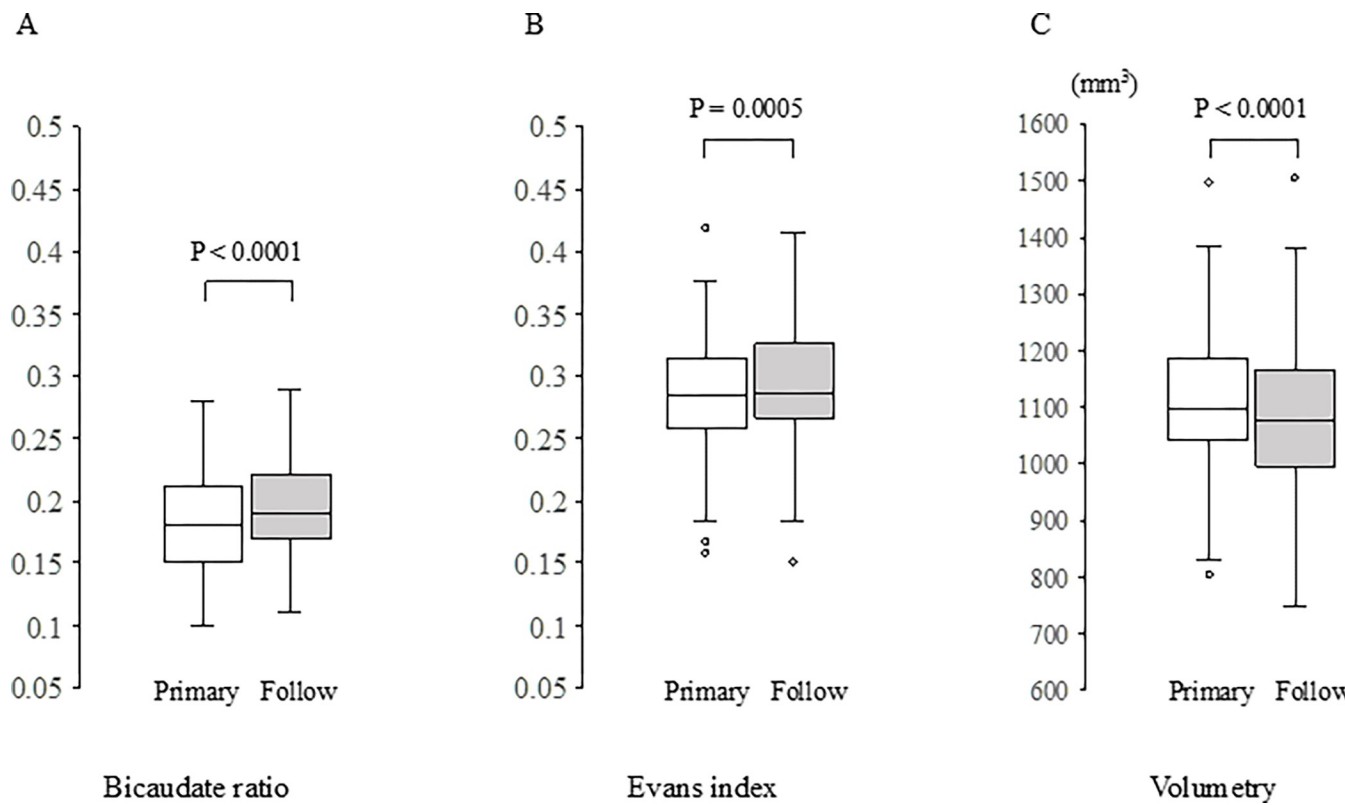

**Fig 2. Comparison of brain volume reduction assessed using head computed tomography between admission and follow-up.** (A) Change in bicaudate ratio. (B) Change in the Evans index. (C) Change in brain volume by volumetry. The Wilcoxon signed-rank test was performed to compare the two groups of paired nonparametric data. *$P < 0.05$.

### Association between brain volume and type of bacteria, bacteremia, APACHE II score, SOFA score, and shock

There was no difference in the degree of decrease in brain volume depending on the type of initiating organism or the presence or absence of bacteremia. The Steel–Dwass test for BCR, EI, and Vo change rates based on infection foci showed no differences in brain volume changes. No significant differences were found between the groups with and without bacteremia in terms of the percentage changes in BCR, EI, and Vo (BCR: $P = 0.2235$, EI: $P = 0.2883$, Vo: $P = 0.1141$). There were significant differences in APACHE II scores ($P = 0.043$), SOFA scores at admission ($P = 0.0387$), and the presence or absence of shock ($P = 0.012$) between groups with and without decreased brain volume for BCR, but not for EI and Vo (Table 2). There were no significant differences in the worst SOFA scores between groups with and without decreased brain volume (Table 2).

### Association between brain volume reduction and ADL

The examination of 58 patients, including those without brain volume reduction, revealed no significant correlation between the rates of change in BCR/EI/Vo and the Katz index. However, when the examination was limited to patients with decreased brain volume, a significant correlation was found between the rates of change in BCR/EI/Vo and the Katz index. The correlation between the rates of changes in the BCR, EI, Vo and Katz index was examined in 38 patients with increased BCR, 35 patients with increased EI, and 46 patients with decreased Vo.

**Table 2. Differences in background and neurological outcomes between patients with and without brain volume reduction.**

| | Bicaudate ratio | | | Evans index | | | Volumetry | | |
|---|---|---|---|---|---|---|---|---|---|
| | Brain Volume Reduction (n = 38) | No Brain Volume Reduction (n = 20) | P-value | Brain Volume Reduction (n = 35) | No Brain Volume Reduction (n = 23) | P-value | Brain Volume Reduction (n = 46) | No Brain Volume Reduction (n = 12) | P-value |
| Age (years) | 78 (68–86) | 75.5 (70–82.5) | 0.9022 | 78 (69–86) | 76 (68–84) | 0.5088 | 76 (70–86) | 76.5 (64.5–82.25) | 0.403 |
| APACHE II score at admission | 23 (19.5–27) | 27 (21.5–35.75) | 0.043 | 24 (21–27) | 24 (17–35) | 0.65 | 24 (21–29) | 24.5 (19.5–31) | 0.9005 |
| SOFA score at admission | 6.5 (5–9.25) | 8 (6.25–10) | 0.0387 | 7 (5–10) | 7 (6–10) | 0.7008 | 7 (5–10) | 8 (6–11.5) | 0.4224 |
| Worst SOFA score (Average hospitalization day 2) | 7 (4.75–10) | 8 (6.25–11.5) | 0.1127 | 7 (5–10) | 8 (6–10) | 0.3220 | 7 (5–10) | 7 (5.25–11.5) | 0.9615 |
| Septic shock, n (%) | 7 (18.4) | 10 (50) | 0.012 | 9 (25.7) | 8 (34.8) | 0.458 | 12 (26.1) | 5 (41.7) | 0.291 |
| Days in the intensive care unit | 8 (4.75–13.25) | 9 (7.25–12) | 0.2064 | 10 (4–14) | 8 (6–10) | 0.6211 | 10 (6–17) | 8 (4.25–11.25) | 0.4077 |
| Duration of intubation (days) | 3 (0–6) | 0.5 (0–5.75) | 0.6678 | 3 (0–9) | 2 (0–5) | 0.611 | 3 (0–12) | 0 (0–5.75) | 0.2192 |
| DIC at admission, n (%) | 7 (18.4) | 7 (35) | 0.1608 | 9 (25.7) | 5 (21.7) | 0.7293 | 10 (21.7) | 4 (33.3) | 0.4032 |
| Katz index at discharge | 2.5 (0–6) | 1.0 (0–2.75) | 0.0623 | 1.0 (0–4) | 3.0 (0–5) | 0.2261 | 1.5 (0–5) | 1.0 (0–2.5) | 0.3816 |
| WBC (×10$^3$/µl) | 10.55 (7.25–15.6) | 11.5 (8.125–17.4) | 0.7373 | 10.6 (6.6–15.5) | 11.2 (8.1–16.2) | 0.4942 | 10 (7.1–13.1) | 14.4 (5.55–16.925) | 0.977 |
| Hemoglobin (g/dl) | 13.2 (11.475–14.5) | 13.05 (11–15.4) | 0.8315 | 12.9 (11.4–14.5) | 13.3 (11.4–15.3) | 0.5888 | 12.8 (11.4–14.4) | 15.05 (11.925–15.825) | 0.0536 |
| Hematocrit (%) | 40.05 (34.475–43.4) | 40.95 (33.1–45.75) | 0.5449 | 39.8 (33.4–43.4) | 40.9 (34.4–44.2) | 0.4892 | 39.9 (34.5–43.4) | 43.5 (38.4–45.85) | 0.0381 |
| Platelet (×10$^4$/µl) | 18.2 (12.0–27.9) | 18.8 (10.7–29.475) | 0.8829 | 18.1 (12–27.9) | 20.3 (11–30.1) | 0.413 | 21.3 (14.1–28.4) | 19.7 (10.575–31.75) | 0.6868 |
| Albumin (g/dl) | 3.3 (2.7–3.9) | 3.25 (2.725–3.675) | 0.5663 | 3.2 (2.7–3.9) | 3.4 (2.9–3.7) | 0.65 | 3.4 (2.7–3.9) | 3.15 (2.55–3.775) | 0.5384 |
| T.Bilirubin (mg/dl) | 0.65 (0.4575–0.985) | 0.675 (0.4–1.53) | 0.9023 | 0.76 (0.49–1.1) | 0.53 (0.38–0.99) | 0.2831 | 0.65 (0.36–0.94) | 0.93 (0.47–2.16) | 0.2376 |
| AST (U/l) | 42 (24–58.25) | 77 (29.5–378.75) | 0.5663 | 51 (26–93) | 34 (22–83) | 0.6221 | 30 (24–53) | 82 (48–378.75) | 0.0502 |
| ALT (U/l) | 24 (13–47) | 34 (18–273) | 0.1301 | 30 (13–74) | 27 (15–50) | 0.8549 | 19 (13–35) | 77 (28.5–176.25) | 0.0307 |
| Na (mEq/l) | 137 (133.75–141.25) | 141.5 (136–146) | 0.0405 | 138 (134–141) | 140 (136–146) | 0.0603 | 140 (136–144) | 139 (136.25–145.5) | 0.4305 |
| K (mEq/l) | 4.3 (3.875–4.8) | 4.4 (3.9–5.275) | 0.4709 | 4.5 (3.9–4.8) | 4.3 (3.8–4.7) | 0.4936 | 4.5 (3.9–4.8) | 4.5 (4.025–5.2) | 0.403 |
| BUN (mg/dl) | 28.25 (18.525–47.325) | 63.7 (31.375–101.25) | 0.0014 | 31.5 (20.5–52.3) | 46.9 (22.5–69) | 0.2459 | 35.5 (22.5–58.5) | 47.7 (23.2–101.25) | 0.1669 |
| Creatinine (mg/dl) | 1.43 (0.91–2.0825) | 1.895 (1.29–3.8875) | 0.0622 | 1.42 (0.98–2.15) | 1.7 (0.98–4.19) | 0.2941 | 1.47 (0.91–2.27) | 1.8 (1.2775–4.14) | 0.1177 |
| Lactate (mmol/l) | 2.75 (1.975–4.075) | 2.95 (1.9–6.8) | 0.6234 | 3.1 (2.1–4.9) | 2.6 (1.4–4) | 0.332 | 2.8 (2.2–4.8) | 5.05 (2.3–7.125) | 0.0757 |
| HCO$_3^-$ (mmol/l) | 20.5 (16.175–23.625) | 18.5 (13.825–23.225) | 0.2624 | 20.3 (16.5–23.5) | 19.6 (11.1–23.5) | 0.8301 | 20.4 (16.1–23.1) | 18.5 (14.475–22.15) | 0.2823 |
| CRP (mg/dl) | 6.55 (0.635–22.2675) | 16.9 (3.79–27.205) | 0.0936 | 13.39 (1.09–25.29) | 9.99 (1.65–26.09) | 0.8239 | 3.12 (0.47–18.93) | 17.78 (6.3–28.86) | 0.0758 |
| Blood glucose level (mg/dl) | 164 (108.75–220.75) | 147 (93.5–348.75) | 0.8829 | 157 (98–220) | 168 (108–289) | 0.2833 | 156 (109–289) | 166.5 (98.5–189.75) | 0.6798 |
| AT3 (%) | 77.5 (64–87.25) | 80.5 (62.5–93.75) | 0.8894 | 72 (60–88) | 83 (69–95) | 0.169 | 84 (68–95) | 80.5 (65.5–89.25) | 0.4255 |
| UA (mg/dl) | 7.2 (5.5–9.5) | 7.95 (5.25–16.05) | 0.2875 | 7.45 (5.425–8.975) | 7.2 (5.5–16.4) | 0.3994 | 7.4 (6.1–9.5) | 7.45 (5.05–18.875) | 0.5711 |
| Fluid balance for 24 after admission | 2278 (1217.5–3420.5) | 2687.5(1325.25–5013.25) | 0.254 | 2326 (1162–3775) | 2258 (1380–3817.25) | 0.8217 | 2238 (1311–3521) | 2825 (1146.25–6404) | 0.2702 |

*(Continued)*

**Table 2.** (Continued)

| | Bicaudate ratio | | | Evans index | | | Volumetry | | |
|---|---|---|---|---|---|---|---|---|---|
| | Brain Volume Reduction (n = 38) | No Brain Volume Reduction (n = 20) | P-value | Brain Volume Reduction (n = 35) | No Brain Volume Reduction (n = 23) | P-value | Brain Volume Reduction (n = 46) | No Brain Volume Reduction (n = 12) | P-value |
| Fluid balance on the day of the final CT | 444(150–623.5) | 398.5 (11.5–561.5) | 0.3644 | 430 (118–607) | 459.5 (133.75–636) | 0.7537 | 444 (140–593) | 419 (92.5–624.75) | 0.9614 |

Abbreviations: *ALT* alanine aminotransferase, *APACHE* Acute Physiology and Chronic Health Evaluation, *AST* aspartate aminotransferase, *AT* antithrombin, *BUN* blood urea nitrogen, *CRP* C-reactive protein, *CT* computed tomography, *DIC* disseminated intravascular coagulation, *SOFA* Sequential Organ Failure Assessment, *UA* uric acid, *WBC* white blood cell.

Nonparametric data are presented as median with interquartile range. Categorical data are presented as n (%). *P*-value <0.05 is considered statistically significant.

A significant correlation was found between the rate of change in Vo and the Katz index ($\rho = -0.3790$, $P = 0.0094$) (Fig 3).

The comparison of demographic characteristics between the two groups—one with and the other without brain volume reduction—showed that the APACHE II score ($P = 0.0430$), the SOFA score ($P = 0.0387$), septic shock ($P = 0.0120$), serum sodium level ($P = 0.0405$), and serum urea nitrogen level ($P = 0.0014$) differed according to the BCR, whereas the serum hematocrit value ($P = 0.0381$) and ALT level ($P = 0.0307$) differed according to Vo (Table 2). Regarding volumetry, the median Katz index for the brain volume reduction group was 1.5 and that for the group with no brain volume reduction was 1.0 (Table 2). However, there was no significant difference between the two groups ($P = 0.3816$). Furthermore, the mean (± standard deviation) of the Katz index was 2.46 ± 2.40 for the brain volume reduction group and 1.58 ± 2.14 for the group without brain volume reduction, indicating high variability.

### Assessment of fluid balance

The in-out fluid balance for 24 h after admission and on the day of the final CT was compared between the two groups with and without brain volume reduction. There were no significant differences between the two groups in BCR, EI, and Vo (Table 2).

## Discussion

This prospective study had several key findings. Based on the head CT findings on admission, we found that brain volume was reduced during the course of sepsis. In addition, in patients with progressive brain volume reduction, there was a significant positive correlation between the rate of brain volume changes and the Katz index, which measures ADL function. Previous reports have shown that some patients have decreased brain volume during the acute phase in sepsis [13,14]. A brain MRI study in patients with sepsis and neurological changes after admission to the ICU showed that approximately 16% of patients had brain volume reduction [14]. In addition, some MRI abnormalities (acute cerebral infarction 22.6%, white matter lesions 16.1%) were observed in half of the patients [13,23,24]. In studies that examined the volume of each part of the brain in patients with cerebral dysfunction associated with sepsis, the most significant reduction in brain volume was found in the white matter of the cerebrum; however, the deep gray matter and cerebellar cortex were relatively unaffected [25–27]. As the present results differed depending on the method used to measure brain volume reduction, it was not possible to examine which part of the brain was reduced in volume. The reason for this could be that BCR and EI evaluate the width between the anterior horns in a flat plane, while Vo

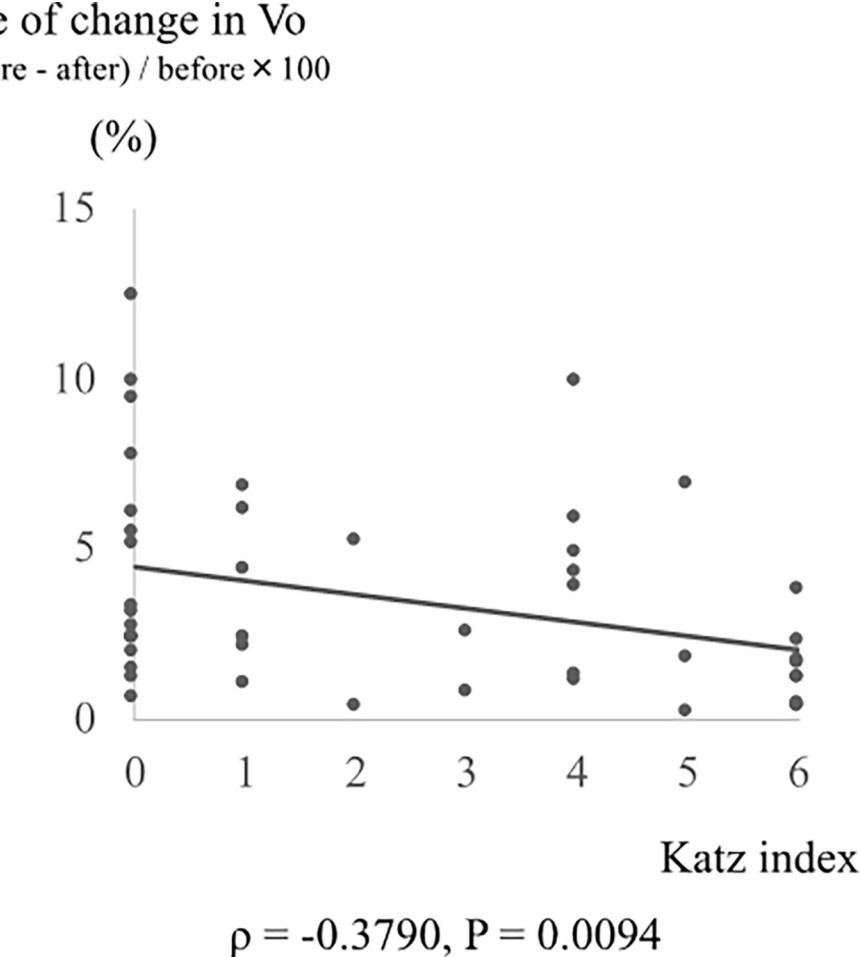

**Fig 3. Association between the degree of brain volume reduction and activities of daily living function.** Correlation between rate of change in the brain volume by volumetry (Vo) and the Katz index. Spearman's rank correlation test was performed to analyze the correlation between the rate of change in Vo and the Katz index. The coefficient of correlation is shown by ρ, *P < 0.05. The reference line shows a positive correlation or tendency.

evaluates the entire brain in a three-dimensional plane. Moreover, in a study of patients with sepsis-associated encephalopathy, a significant decrease in brain volume was observed in coma patients [28]. However, none of those previous reports studied the time course starting from the acute phase [13,14], and most were comparisons with healthy volunteers [24,26,29,30]. Although brain MRI is the best method for evaluating the brain during sepsis [31], given the extreme difficulty of performing frequent MRI during the systemic management of sepsis, we used head CT in this study.

We also assessed whether fluid balance affected the reduction in brain volume. There were no significant differences in fluid balance for 24 h after admission and on the day of the final CT between the groups with and without brain volume reduction. Specifically, brain volume reduction had no causal relationship with fluid balance or cerebral edema.

Brain volume reduction in the early phase in patients with sepsis could indicate a clinical predisposition for developing non-favorable ADL function, with increased risk of post-intensive care syndrome. The Katz index was used to evaluate ADL performance in this study. This index is widely used in clinical studies to assess hospital-associated disability, meaning a

decrease in ADL function associated with hospitalization [22]. However, we did not evaluate long-term ADL function; further studies are needed to identify whether brain volume reduction could be a novel target for sepsis-associated encephalopathy. A multicenter study reported that approximately half of the patients with sepsis had sepsis-associated encephalopathy [32]; thus, sepsis-associated encephalopathy might be related to the mechanism of brain volume reduction.

Brain volume reduction has been reported to occur between the 9th and 12th days of onset in patients with pediatric acute encephalopathy [33] and to begin on the 3rd day after head trauma [34]. Even in the acute phase of ICU treatment, various factors such as sepsis may contribute to brain volume reduction. Brain atrophy is an irreversible change [35], but the design of this study is insufficient to conclude whether the volume reduction observed corresponded to brain atrophy, as this variable was not examined over the long term. Especially in sepsis, it is well known that consciousness disorder is complicated, and it is considered to be a diffuse brain disorder resulting from systemic inflammatory response syndrome secondary to infection [36], although the exact mechanism has been unclear. In recent years, such a condition has been called sepsis-associated encephalopathy and is regarded as an important pathological condition associated not only to survival rates but also to the rate of reintegration into society [6,7]. Therefore, to improve the outcome of patients with sepsis, it is necessary not only to reduce the mortality rate but also to improve this rate of reintegration, including long-term neurological outcomes such as ADL function. Importantly, the present study revealed that patients with sepsis and brain volume reduction from the initial stage of ICU admission had decreased ADL function relative to neurological outcomes at discharge or transfer. By clarifying the mechanism of brain volume reduction in sepsis, we believe that a new treatment strategy for improving the rate of reintegration into society may be developed in the future.

The present study investigated possible reasons for brain volume reduction during the acute phase of sepsis, assuming an influence of the severity of sepsis, bacterial species, and cerebral ischemia caused by hypotension or hypoxia during management (secondary brain damage due to cerebral ischemia). Regarding the severity of illness, a statistical analysis of the relationship between SOFA scores (as a metric for the severity of sepsis at admission and at worst during the disease course) and hypoxia or hypotension (because the SOFA score includes items for hypotension and hypoxia) yielded a significant relationship between SOFA scores at admission and brain volume reduction for BCR, not the other index. Moreover, we examined whether the type of bacteria and the presence of bacteremia were related to decreases in brain volume. Orhun et al. [14] reported that significantly more patients with sepsis-induced brain dysfunction with brain lesions had positive blood culture results; however, no significant differences in brain atrophy were observed in this study. Further, various effects could also be ruled out, such as direct effects on brain cells by systemic inflammatory mediators that occur during sepsis [37] and volume reduction due to hyperglycemia, inflammatory cell infiltration, and cell phagocytosis in ischemic brain tissue [38]. However, these effects have not been clarified in the present study. A more detailed analysis of ICU data is needed in the future, including the measurement of inflammatory mediators (such as cytokines), the duration of hypotension and hypoxia, and sustained hyperglycemia.

There are some limitations to this study. The first is the absence of a control group. It is unclear whether the results were associated with sepsis-specific decreased brain volume or other medical conditions (such as hypotension and hypoxia). It would have been beneficial to include a control group without sepsis. Second, the median age of patients in this study was 79 years and, therefore, the included patients were elderly. The specific characteristics of sepsis in older patients may have influenced the results. Therefore, whether the results can be generalized to younger individuals warrants further investigation. Third, the number of patients was

limited in this single-center study, and the most severe patients, who died early (n = 19), were not examined. In addition, repeated head CT is burdensome for patients, and a large number of patients were unable to provide consent, making the sample size of this study small. Fourth, the evaluation period was short. The median evaluation time for brain volume was the 13th day of hospitalization, and the median evaluation time for ADL was the 20th day of hospitalization. Further studies are needed to determine if the acute effects are consistent with long-term ADL function after intensive care. Fifth, the changes in brain volume reduction in our study were small, and the implications of these changes may have been underexamined, such as by not examining long-term ADL performance. Specifically, it has not been possible to examine whether this volume reduction was organic and related to long-term outcomes. In the future, it is necessary to examine the exact location and persistence of the brain volume reduction in the long term. Sixth, it is difficult to identify the time of onset of sepsis [39]. Therefore, as our reference date was the time of admission, there is a possibility that the evaluation date of head CT may vary. Finally, the assessment of brain volume reduction was assessed from head CT instead of brain MRI. In the acute phase of critically ill patients, their condition is often unstable, and it is difficult to safely perform MRI at the time of admission and follow-up. Therefore, we used head CT instead, for the reasons previously explained.

## Conclusion

Brain volume reduction occurred in the acute phase of sepsis in 55%–79% of elderly patients. Brain volume reduction in the acute phase of sepsis was associated with poor ADL function, although the clinical significance of this phenomenon remains unclear. Future studies aimed at clarifying the underlying causes and possible countermeasures will help improve not only survival but also long-term ADL function in patients with sepsis.

## Supporting information

**S1 Data.**
(XLSX)

## Acknowledgments

We thank Kumiko Obayashi (research assistant).

## Author Contributions

**Conceptualization:** Toru Hosokawa, Kosaku Kinoshita.

**Data curation:** Toru Hosokawa, Shingo Ihara, Katsuhiro Nakagawa, Umefumi Iguchi.

**Formal analysis:** Toru Hosokawa, Shingo Ihara.

**Investigation:** Toru Hosokawa, Shingo Ihara, Katsuhiro Nakagawa, Umefumi Iguchi, Marina Hirabayashi, Tomokazu Mutoh, Nami Sawada, Tsukasa Kuwana.

**Methodology:** Toru Hosokawa, Kosaku Kinoshita, Shingo Ihara, Nami Sawada, Tsukasa Kuwana, Junko Yamaguchi.

**Project administration:** Toru Hosokawa, Kosaku Kinoshita, Tsukasa Kuwana, Junko Yamaguchi.

**Supervision:** Kosaku Kinoshita, Tsukasa Kuwana, Junko Yamaguchi.

**Validation:** Toru Hosokawa, Kosaku Kinoshita, Umefumi Iguchi, Marina Hirabayashi, Tomokazu Mutoh, Nami Sawada, Tsukasa Kuwana, Junko Yamaguchi.

**Visualization:** Toru Hosokawa.

**Writing – original draft:** Toru Hosokawa.

**Writing – review & editing:** Toru Hosokawa, Kosaku Kinoshita, Junko Yamaguchi.

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
