## [Decision Letter · Decision Letter 0]

31 Jan 2023

PONE-D-22-33962Relationship between brain volume reduction during the acute phase of sepsis and activities of daily living in elderly patients: a prospective cohort studyPLOS ONE

Dear Dr. Kinoshita,

Thank you for submitting your manuscript to PLOS ONE. After careful consideration, we feel that it has merit but does not fully meet PLOS ONE’s publication criteria as it currently stands. Therefore, we invite you to submit a revised version of the manuscript that addresses the points raised during the review process.

We look forward to receiving your revised manuscript.

Kind regards,

Dong Wook Jekarl

Academic Editor

PLOS ONE

Journal Requirements:

Reviewers' comments:

Reviewer's Responses to Questions

**Comments to the Author**

1. Is the manuscript technically sound, and do the data support the conclusions?

Reviewer #1: Partly

Reviewer #2: No

Reviewer #3: Partly

2. Has the statistical analysis been performed appropriately and rigorously? 

Reviewer #1: Yes

Reviewer #2: Yes

Reviewer #3: Yes

3. Have the authors made all data underlying the findings in their manuscript fully available?

Reviewer #1: Yes

Reviewer #2: Yes

Reviewer #3: Yes

4. Is the manuscript presented in an intelligible fashion and written in standard English?

Reviewer #1: Yes

Reviewer #2: Yes

Reviewer #3: Yes

5. Review Comments to the Author

Reviewer #1: Toru Hosokawa et al. showed the association between brain volume reduction and performance of the activities of daily living via the Katz index. Overall, this is a concise study with a limited number of participants with sepsis or septic shock. There are some points that the authors need to revise.

(Major)

P5 Ln 83 Blood test for what? In Line 83, authors described that blood tests were performed several times, but in Line132, only the results at the time of admission were used for examination.

So, are the biochemical test results used throughout the result (Table 1) at the time of admission?

Wouldn't it be more desirable to process the statistics in Table 2 using the biochemical results at the time of sepsis diagnosis or CT scan performed? It is better to describe when the result of the biochemical test (Table 2) was obtained.

P6 Ln91-93 What does it mean that “samples from the suspected site of infection, diagnostic imaging, and examination data were collected, ----- and bacterial culture results”? Please clarify the statement.

P14 Table1(B) Clarify the "others” (n = 3)

P14 Table1(C) It would be better to add more combined information for site of infection and bacteria identification results. In addition, if there was “no pathogen”, but is suspected of infection, additional description is required, and the reason for why no pathogen is also required(e.g., Improper specimen, use of antibiotics prior to specimen collection). Since it seems meaningless to present the number of simple “pure gram negative” and “pure gram positive” as Table 1(c), it is better to describe a table that show which bacteria were identified in which part by combining 1(B) and 1(C).

P17 Ln231-236 There is no description in the discussion about why some demographic characteristics show differences according to BCR and VO. The possible causes for this difference should be described in the Discussion section.

Was there any association with brain volume reduction in terms of bacterial isolation from any infection site? Or was there no relationship between bacteremia and brain volume reduction?

(Minor)

Change "P" in P-values to italic.

Typographical error in Table 2 (p value for platelet in Bicaudate ratio

Reviewer #2: Sepsis - septic shock includes a very heterogenous cohort with great variation in disease severity. Althoug of the prospective design of this study, the number of patients is relative low for study of 2 subcohorts.

Reviewer #3: This study presented a very interesting topic by studying the relationship between the occurrence of a decrease in brain volume and ADL in patients with sepsis. In addition, the analysis of the results using CT at two time point showed the novelty of this study. However, for the completeness of this study, some additional descriptions or more detailed explanations are needed.

First, it seems necessary to explain the relationship between sepsis and brain volume, the main content of this study. What factors of sepsis are involved in the reduction of brain volume will help to understand the various measurement results presented in this study.

Other detailed questions are as follows.

Line 30: It can be misunderstood as an expression that the brain volume has decreased by 60 to 79%. Please clarify this part.

Line 83, 84: The timing of the blood test is described. Results related to this part are needed.

Line 119: “Scoring two or more” can be misunderstood as an expression of higher availability. Please clarify this part.

Line 122: What does sample mean? Please clarify this part.

Line 129: A comma (,) to be needed between hematocrit and albumin.

Line 177: In Table 1, the mean value of ADL is 1, but in Figure 3, it seems to be higher. Then, did the group without a change in the brain volume show a very low Katz index level? Table 3 describes Katz indexes of groups with and without changes in brain volume. Considering these figures, the relationship between the change in Vo and ADL appears to be less relevant, and it is recommended to add an explanation.

Line 218-219: It is understood that significant correlation was seen only in the patient group where the brain volume decreased. Please clarify this sentence.

Line 232: In the patient group with reduced brain volume, the values of APACHE II and SOFA score are low, so the relationship between sepsis severity and brain volume reduction seems to be low. The percentage of septic shock is also low. It seems necessary to add an explanation for this part.

6. PLOS authors have the option to publish the peer review history of their article (what does this mean?). If published, this will include your full peer review and any attached files.

Reviewer #1: No

Reviewer #2: No

Reviewer #3: No

---

## [Author Response · Author response to Decision Letter 0]

1 Apr 2023

April 1, 2023

Response to the reviewer comments

Thank you for your kind comments about our article. We have revised the manuscript in accordance with your suggestions and those of the reviewers. The revised sections are highlighted (below and in the main manuscript file), and we have provided point-by-point responses below.

Comments to the Author:

Reviewer #1:

Comment 1. P5 Ln 83 Blood test for what? In Line 83, authors described that blood tests were performed several times, but in Line132, only the results at the time of admission were used for examination. So, are the biochemical test results used throughout the result (Table 1) at the time of admission? Wouldn't it be more desirable to process the statistics in Table 2 using the biochemical results at the time of sepsis diagnosis or CT scan performed? It is better to describe when the result of the biochemical test (Table 2) was obtained.

Response: (Reviewer #1 comment 1)

Thank you for this important comment. We performed blood tests on admission (day 0) and on days 7, 14, 21, and 28. These periodic tests were performed routinely to determine the course of treatment for sepsis. We used these results to calculate the daily Sequential Organ Failure Assessment (SOFA) scores in combination with the daily biochemical test results. We then used the daily SOFA scores to identify the worst SOFA score. We have revised Table 2 with the addition of the worst SOFA scores. In addition, this study examined the decreases in brain volume during hospitalization in the group diagnosed with sepsis at the time of admission. Therefore, the results at admission were the same as those at the time of sepsis diagnosis. Because we did not stratify the patients during the course of treatment, we only presented the data at admission (i.e., at the time of sepsis diagnosis). Therefore, the statistical analysis in Table 2 was also performed using the data from the time of admission. As you have indicated, the timing of obtaining the biochemical test results (Table 2) was unclear, so we have made the following additions to the text.

The following statement has been added to the methods section (P5 Ln87–90).

These periodic tests were routinely performed to determine the course of sepsis treatment. We used these results to calculate the daily Sequential Organ Failure Assessment (SOFA) scores. 

The following statement has been added to the methods section (P6 Ln92–93).

The diagnosis of sepsis was based on vital signs and blood test findings on admission.

The following statement has been added to Table 1 (P16 Ln207–208).

The biochemical data in this table are based on data at the time of admission (i.e., at the time of sepsis diagnosis).

Comment 2. P6 Ln91-93 What does it mean that “samples from the suspected site of infection, diagnostic imaging, and examination data were collected, ----- and bacterial culture results”? Please clarify the statement.

Response: (Reviewer #1 comment 2)

As you have indicated, the wording was unclear. The following has been added to the methods section (P6 Ln96–103).

To confirm the diagnosis of infectious illness, blood and bacterial cultures of specimens taken from organs considered to be the focus of infection based on clinical findings, imaging findings, and laboratory data were assessed. Although all patients met the Sepsis-3 criteria and clinical criteria for infectious illness, a subgroup of patients with negative bacterial culture results were defined as cases of unknown origin. The group classification was based on whether the bacterial cultures of specimens taken from the sites considered to be of septic focus were positive. 

Comment 3. P14 Table1(B) Clarify the "others” (n = 3).

Response: (Reviewer #1 comment 3)

The category of “others” included the following three cases:

1. periapical dental infection

2. infective endocarditis

3. bacteremia with unknown site of infection

The following has been added to the results section (P11 Ln190-191) and to Table 1 (P16 Ln208–210).

The “other” category included one case each of periapical dental infection, infectious endocarditis, and bacteremia with unknown foci.

Comment 4. P14 Table1(C) It would be better to add more combined information for site of infection and bacteria identification results. In addition, if there was “no pathogen”, but is suspected of infection, additional description is required, and the reason for why no pathogen is also required(e.g., Improper specimen, use of antibiotics prior to specimen collection). Since it seems meaningless to present the number of simple “pure gram negative” and “pure gram positive” as Table 1(c), it is better to describe a table that show which bacteria were identified in which part by combining 1(B) and 1(C). 

Response: (Reviewer #1 comment 4)

Table 1B and C were combined to show the combination of infection sites and bacterial identification results, and the table was revised (Table 1B).

Please find the new table 1B in the article body.

The reasons for a determination of “no pathogen” have been added to the results section as follows (P11 Ln195–P12 Ln197).

Pathogens could not be identified in 5 (8.6%) patients (Table 1B) because the specimens were insufficient for analysis in three cases and antimicrobial agents were administered before specimen collection in two cases.

Comment 5. P17 Ln231-236 There is no description in the discussion about why some demographic characteristics show differences according to BCR and VO. The possible causes for this difference should be described in the Discussion section.

Response: (Reviewer #1 comment 5)

As you have pointed out, different measurement methods (i.e., the Bicaudate ratio [BCR], Evans index [EI], and Volumetry [Vo], produced different results. This could be due to differences in their measurement methods and characteristics. The measurement methods and characteristics have been described in the text on P7 L111–123. We have added the following to make this clear.

The following statement has also been added in the discussion section (P27 Ln297–P28 Ln301).

As the present results differed depending on the method used to measure brain volume reduction, it was not possible to examine which part of the brain was reduced in volume. The reason for this could be that BCR and EI evaluate the width between the anterior horns in a flat plane, while Vo evaluates the entire brain in a three-dimensional plane.

Comment 6. Was there any association with brain volume reduction in terms of bacterial isolation from any infection site? Or was there no relationship between bacteremia and brain volume reduction?

Response: (Reviewer #1 comment 6)

Please refer to the revised Table 1B for the bacterial species detected at the sites of infection. As you have indicated, we examined whether the type of bacteria and the presence of bacteremia were related to decreases in brain volume, but no associations were observed.

The following has been added to the results section (P17 Ln228–235).

Association between brain volume and type of bacteria, bacteremia, APACHE II score, SOFA score, and shock

There was no difference in the degree of decrease in brain volume depending on the type of initiating organism or the presence or absence of bacteremia. The Steel–Dwass test for BCR, EI, and Vo change rates based on infection foci showed no differences in brain volume changes. No significant differences were found between the groups with and without bacteremia in terms of the percentage changes in BCR, EI, and Vo (BCR: P = 0.2235, EI: P = 0.2883, Vo: P = 0.1141). 

The following statement has also been added to the discussion section (P30 Ln352–P31 Ln356).

Moreover, we examined whether the type of bacteria and the presence of bacteremia were related to decreases in brain volume. Orhun et al. [14] reported that significantly more patients with sepsis-induced brain dysfunction with brain lesions had positive blood culture results; however, no significant differences in brain atrophy were observed in this study.

Comment 7. Change "P" in P-values to italic.

Response: (Reviewer #1 comment 7)

Thank you for this helpful comment. The “P” in P-values has been changed to italics in the revised manuscript.

Comment 8. Typographical error in Table 2 (p value for platelet in Bicaudate ratio).

Response: (Reviewer #1 comment 8)

Thank you for pointing this out. We have corrected the P-value for platelets in the bicaudate ratio from 8829 to 0.8829.

Reviewer #2:

Comment 1. Sepsis - septic shock includes a very heterogenous cohort with great variation in disease severity. Althoug of the prospective design of this study, the number of patients is relative low for study of 2 subcohorts.

Response: (Reviewer #2 comment 1)

In recent years, sepsis-related encephalopathy, a complication of sepsis-induced brain dysfunction, has gained importance as a condition that affects not only survival but also the rate of return to society. Sepsis-related encephalopathy is a diffuse brain disorder caused by a systemic inflammatory response that occurs due to sepsis, and mitochondrial dysfunction, microglial activation, and vascular endothelial cell activation have been proposed, but no specific mechanisms have been identified. Therefore, we decided to examine decreases in brain volume due to organic brain dysfunction, because we believe that sepsis with brain dysfunction could be caused by organic brain damage and that systemic inflammation caused by sepsis could be involved. The subject of this study was not sepsis-related encephalopathy, but rather patients diagnosed with sepsis at the time of admission.

As you have noted, the relationship between decreases in brain volume and the severity of shock and other symptoms is important. With regard to the severity of illness on admission, there were significant differences in APACHE II scores (P = 0.043), SOFA scores at admission (P = 0.0387), and the presence or absence of shock (P = 0.012) between groups with and without decreased brain volume for BCR, but not for EI and Vo (Table 2).

Moreover, hypotension and hypoxia can cause cerebral ischemia, which could contribute to brain volume reduction. Therefore, we reexamined the relationship between the worst SOFA score (in each patient) and brain volume reduction during the study period. Hypotension and hypoxemia are components of the SOFA score; however, no significant associations were demonstrated in the present study. Perhaps the degree and duration of hypotension and hypoxemia could have had an effect, but this was not examined.

In addition, as you have pointed out, the sample size of this study was small. However, repeated head CT is burdensome for patients, and because there were a number of patients who did not give consent, this was unavoidable.

We have added the results of our examination of the worst SOFA score to Table 2 (P20–26). 

We revised the Table 2. Please refer new table 2.

The following has been added to the results (P17 Ln228–229).

Association between brain volume and type of bacteria, bacteremia, APACHE II score, SOFA score, and shock

and P17 Ln235–240

There were significant differences in APACHE II scores (P = 0.043), SOFA scores at admission (P = 0.0387), and the presence or absence of shock (P = 0.012) between groups with and without decreased brain volume for BCR, but not for EI and Vo (Table 2). There were no significant differences in the worst SOFA scores between groups with and without decreased brain volume (Table 2).

The following statement has also been added to the discussion section (P30 Ln344–P31 Ln363).

The present study investigated possible reasons for brain volume reduction during the acute phase of sepsis, assuming an influence of the severity of sepsis, bacterial species, and cerebral ischemia caused by hypotension or hypoxia during management (secondary brain damage due to cerebral ischemia). Regarding the severity of illness, a statistical analysis of the relationship between SOFA scores (as a metric for the severity of sepsis at admission and at worst during the disease course) and hypoxia or hypotension (because the SOFA score includes items for hypotension and hypoxia) yielded a significant relationship between SOFA scores at admission and brain volume reduction for BCR, not the other index. Moreover, we examined whether the type of bacteria and the presence of bacteremia were related to decreases in brain volume. Orhun et al. [14] reported that significantly more patients with sepsis-induced brain dysfunction with brain lesions had positive blood culture results; however, no significant differences in brain atrophy were observed in this study. Further, various effects could also be ruled out, such as direct effects on brain cells by systemic inflammatory mediators that occur during sepsis [37] and volume reduction due to hyperglycemia, inflammatory cell infiltration, and cell phagocytosis in ischemic brain tissue [38]. However, these effects have not been clarified in the present study. A more detailed analysis of ICU data is needed in the future, including the measurement of inflammatory mediators (such as cytokines), the duration of hypotension and hypoxia, and sustained hyperglycemia.

The following statement has also been added to the discussion section (P32 Ln372–374).

In addition, repeated head CT is burdensome for patients, and a large number of patients were unable to provide consent, making the sample size of this study small.

Reviewer #3:

Comment 1. This study presented a very interesting topic by studying the relationship between the occurrence of a decrease in brain volume and ADL in patients with sepsis. In addition, the analysis of the results using CT at two time point showed the novelty of this study. However, for the completeness of this study, some additional descriptions or more detailed explanations are needed.

First, it seems necessary to explain the relationship between sepsis and brain volume, the main content of this study. What factors of sepsis are involved in the reduction of brain volume will help to understand the various measurement results presented in this study.

Other detailed questions are as follows.

Response: (Reviewer #3 comment 1)

As you have pointed out, it is very important to ask what aspects of the pathophysiology of sepsis are associated with reduced brain volume. The mechanisms of brain volume reduction during the acute phase of sepsis were investigated in this study, assuming an influence of the severity of sepsis, bacterial species, and cerebral ischemia caused by hypotension or hypoxia during disease management (secondary brain damage due to cerebral ischemia). Then, a statistical analysis of the relationship between SOFA scores as a metric of sepsis severity at admission and at worst during the course of the disease and hypoxia or hypotension (because the SOFA score includes items for hypotension and hypoxia) yielded a significant relationship between the SOFA score at admission and brain volume reduction for BCR. Moreover, we examined whether the type of bacteria and the presence of bacteremia were related to decreases in brain volume, but no significant differences were observed in brain atrophy. As the other factor, various effects could also be ruled out, such as direct effects on brain cells by systemic inflammatory mediators that occur during sepsis (Deutschman CS, Tracey KJ. Sepsis: current dogma and new perspectives. Immunity. 2014;40: 463-475), and volume reduction due to hyperglycemia, inflammatory cell infiltration, and cell phagocytosis in ischemic brain tissue (Lin B, Ginsberg MD, Busto R, Li L. Hyperglycemia triggers massive neutrophil deposition in brain following transient ischemia in rats. Neurosci Lett. 2000;278: 1-4). However, these effects have not been clarified in the present study. More detailed analysis of ICU data is needed in the future, including the measurement of inflammatory mediators (such as cytokines), the duration of hypotension and hypoxia, and sustained hyperglycemia.

The following statement has been added to the discussion section (P30 Ln344–P31 Ln363).

The present study investigated possible reasons for brain volume reduction during the acute phase of sepsis, assuming an influence of the severity of sepsis, bacterial species, and cerebral ischemia caused by hypotension or hypoxia during management (secondary brain damage due to cerebral ischemia). Regarding the severity of illness, a statistical analysis of the relationship between SOFA scores (as a metric for the severity of sepsis at admission and at worst during the disease course) and hypoxia or hypotension (because the SOFA score includes items for hypotension and hypoxia) yielded a significant relationship between SOFA scores at admission and brain volume reduction for BCR, not the other index. Moreover, we examined whether the type of bacteria and the presence of bacteremia were related to decreases in brain volume. Orhun et al. [14] reported that significantly more patients with sepsis-induced brain dysfunction with brain lesions had positive blood culture results; however, no significant differences in brain atrophy were observed in this study. Further, various effects could also be ruled out, such as direct effects on brain cells by systemic inflammatory mediators that occur during sepsis [37] and volume reduction due to hyperglycemia, inflammatory cell infiltration, and cell phagocytosis in ischemic brain tissue [38]. However, these effects have not been clarified in the present study. A more detailed analysis of ICU data is needed in the future, including the measurement of inflammatory mediators (such as cytokines), the duration of hypotension and hypoxia, and sustained hyperglycemia.

Comment 2. Line 30: It can be misunderstood as an expression that the brain volume has decreased by 60 to 79%. Please clarify this part.

Response: (Reviewer #3 comment 2)

The following modifications have been made (P2 Ln33–34).

In the acute phase of sepsis in this sample of older patients, 60–79% of patients showed decreased brain volumes.

Comment 3. Line 83, 84: The timing of the blood test is described. Results related to this part are needed.

Response: (Reviewer #3 comment 3)

Blood tests after admission were routinely performed to determine the effects of treatment for sepsis. The results were used to calculate the worst SOFA score (for each patient) as an assessment of the severity of sepsis during the course of treatment. The results were used to examine associations with brain volume reduction; this has been added to Table 2.

We have added the results of our examinations of the SOFA scores at worst to Table 2 (P20–26).

Comment 4. Line 119: “Scoring two or more” can be misunderstood as an expression of higher availability. Please clarify this part.

Response: (Reviewer #3 comment 4)

The following modifications have been made (P8 Ln129–130).

(i.e., requiring assistance with at least two of the following: bathing, dressing, toilet use, transferring, continence, and eating)

Comment 5. Line 122: What does sample mean? Please clarify this part.

Response: (Reviewer #3 comment 5)

Thank you for this important comment. As you have pointed out, we have modified the text as follows so that the contents of the “sample” could be understood (P8 Ln131–132).

To examine the association between brain volume reduction and ADL, the entire study population, including patients with brain volume reduction, was analyzed.

Comment 6. Line 129: A comma (,) to be needed between hematocrit and albumin.

Response: (Reviewer #3 comment 6)

Thank you for this important comment. This has been corrected as indicated (P8 Ln138).

Comment 7. Line 177: In Table 1, the mean value of ADL is 1, but in Figure 3, it seems to be higher. Then, did the group without a change in the brain volume show a very low Katz index level? Table 3 describes Katz indexes of groups with and without changes in brain volume. Considering these figures, the relationship between the change in Vo and ADL appears to be less relevant, and it is recommended to add an explanation.

Response: (Reviewer #3 comment 7)

First, an error was made in the values in Table 2, which has been corrected. In the volumetry, the Katz index for the brain volume reduction group was 1.5 (0–5), not 3 (0–4). Further, as you have noted, in the volumetry, the median Katz index for the brain volume reduction group was 1.5 and the median Katz index for the group without brain volume reduction group was 1 (Table 2). However, there was no significant difference between the two groups (P-value = 0.3816). Furthermore, when the mean ± standard deviation of the Katz index were calculated, these were 2.46 ± 2.40 for the brain volume reduction group and 1.58 ± 2.14 for the group with no brain volume reduction, indicating that there was a large variability.

The following statement has been added to the results section (P19 Ln265–270).

Regarding volumetry, the median Katz index for the brain volume reduction group was 1.5 and that for the group with no brain volume reduction was 1.0 (Table 2). However, there was no significant difference between the two groups (P = 0.3816). Furthermore, the mean (± standard deviation) of the Katz index was 2.46 ± 2.40 for the brain volume reduction group and 1.58 ± 2.14 for the group without brain volume reduction, indicating high variability.

Comment 8. Line 218-219: It is understood that significant correlation was seen only in the patient group where the brain volume decreased. Please clarify this sentence.

Response: (Reviewer #3 comment 8)

As you have indicated, the text was not clear, thus it has been corrected as follows (P18 Ln243–251).

The examination of 58 patients, including those without brain volume reduction, revealed no significant correlation between the rates of change in BCR/EI/Vo and the Katz index. However, when the examination was limited to patients with decreased brain volume, a significant correlation was found between the rates of change in BCR/EI/Vo and the Katz index. The correlation between the rates of changes in the BCR, EI, Vo and Katz index was examined in 38 patients with increased BCR, 35 patients with increased EI, and 46 patients with decreased Vo. A significant correlation was found between the rate of change in Vo and the Katz index (ρ = −0.3790, P = 0.0094) (Fig 3).

Comment 9. Line 232: In the patient group with reduced brain volume, the values of APACHE II and SOFA score are low, so the relationship between sepsis severity and brain volume reduction seems to be low. The percentage of septic shock is also low. It seems necessary to add an explanation for this part.

Response: (Reviewer #3 comment 9)

With regard to the severity of illness at admission, there were significant differences in APACHE II scores (P = 0.043), SOFA scores at admission (P = 0.0387), and the presence or absence of shock (P = 0.012) between groups with and without decreased brain volume for BCR (Table 2). However, the relationship between the worst SOFA score (for each patient) and brain volume reduction during the course of sepsis was re-examined, and no significant association was found. Please also see the response to Comment 1 from Reviewer #2. We do consider the possibility that perhaps the degree and duration of hypotension and hypoxemia, as well as hyperglycemia, may have influenced the results, but this was not examined in this study and is a topic for future work.

As per our response to Comment 1 from Reviewer #2, we have added the results of our examination of the SOFA scores at worst to Table 2 (P20–26).

As per our response to Comment 1 from Reviewer #2, we have added to the following sentence to the discussion section (P30 Ln344-–P31 Ln363).

The present study investigated possible reasons for brain volume reduction during the acute phase of sepsis, assuming an influence of the severity of sepsis, bacterial species, and cerebral ischemia caused by hypotension or hypoxia during management (secondary brain damage due to cerebral ischemia). Regarding the severity of illness, a statistical analysis of the relationship between SOFA scores (as a metric for the severity of sepsis at admission and at worst during the disease course) and hypoxia or hypotension (because the SOFA score includes items for hypotension and hypoxia) yielded a significant relationship between SOFA scores at admission and brain volume reduction for BCR, not the other index. Moreover, we examined whether the type of bacteria and the presence of bacteremia were related to decreases in brain volume. Orhun et al. [14] reported that significantly more patients with sepsis-induced brain dysfunction with brain lesions had positive blood culture results; however, no significant differences in brain atrophy were observed in this study. Further, various effects could also be ruled out, such as direct effects on brain cells by systemic inflammatory mediators that occur during sepsis [37] and volume reduction due to hyperglycemia, inflammatory cell infiltration, and cell phagocytosis in ischemic brain tissue [38]. However, these effects have not been clarified in the present study. A more detailed analysis of ICU data is needed in the future, including the measurement of inflammatory mediators (such as cytokines), the duration of hypotension and hypoxia, and sustained hyperglycemia.

---

## [Decision Letter · Decision Letter 1]

11 Apr 2023

Relationship between brain volume reduction during the acute phase of sepsis and activities of daily living in elderly patients: a prospective cohort study

PONE-D-22-33962R1

Dear Dr. Kinoshita,

We’re pleased to inform you that your manuscript has been judged scientifically suitable for publication and will be formally accepted for publication once it meets all outstanding technical requirements.

Kind regards,

Dong Wook Jekarl

Academic Editor

PLOS ONE

Additional Editor Comments (optional):

Reviewers' comments:

Reviewer's Responses to Questions

**Comments to the Author**

1. If the authors have adequately addressed your comments raised in a previous round of review and you feel that this manuscript is now acceptable for publication, you may indicate that here to bypass the “Comments to the Author” section, enter your conflict of interest statement in the “Confidential to Editor” section, and submit your "Accept" recommendation.

Reviewer #3: All comments have been addressed

Reviewer #4: (No Response)

2. Is the manuscript technically sound, and do the data support the conclusions?

Reviewer #3: Yes

Reviewer #4: (No Response)

3. Has the statistical analysis been performed appropriately and rigorously? 

Reviewer #3: Yes

Reviewer #4: (No Response)

4. Have the authors made all data underlying the findings in their manuscript fully available?

Reviewer #3: Yes

Reviewer #4: (No Response)

5. Is the manuscript presented in an intelligible fashion and written in standard English?

Reviewer #3: Yes

Reviewer #4: (No Response)

6. Review Comments to the Author

Reviewer #3: (No Response)

Reviewer #4: (No Response)

7. PLOS authors have the option to publish the peer review history of their article (what does this mean?). If published, this will include your full peer review and any attached files.

Reviewer #3: No

Reviewer #4: No

---

## [Editor Report · Acceptance letter]

8 May 2023

PONE-D-22-33962R1 

Relationship between brain volume reduction during the acute phase of sepsis and activities of daily living in elderly patients: a prospective cohort study 

Dear Dr. Kinoshita:

I'm pleased to inform you that your manuscript has been deemed suitable for publication in PLOS ONE. Congratulations! Your manuscript is now with our production department. 

Kind regards, 

on behalf of

Dr. Dong Wook Jekarl 

Academic Editor

PLOS ONE